# DIET-SNN: A Low-Latency Spiking Neural Network with Direct Input Encoding & Leakage and Threshold Optimization

## Abstract

Bio-inspired spiking neural networks (SNNs), operating with asynchronous binary signals (or spikes) distributed over time, can potentially lead to greater computational efficiency on event-driven hardware. The state-of-the-art SNNs suffer from high inference latency, resulting from inefficient input encoding, and sub-optimal settings of the neuron parameters (firing threshold, and membrane leak). We propose DIET-SNN, a low-latency deep spiking network that is trained with gradient descent to optimize the membrane leak and the firing threshold along with other network parameters (weights). The membrane leak and threshold for each layer of the SNN are optimized with end-to-end backpropagation to achieve competitive accuracy at reduced latency. The analog pixel values of an image are directly applied to the input layer of DIET-SNN without the need to convert to spike-train. The first convolutional layer is trained to convert inputs into spikes where leaky-integrate-and-fire (LIF) neurons integrate the weighted inputs and generate an output spike when the membrane potential crosses the trained firing threshold. The trained membrane leak controls the flow of input information and attenuates irrelevant inputs to increase the activation sparsity in the convolutional and dense layers of the network. The reduced latency combined with high activation sparsity provides large improvements in computational efficiency. We evaluate DIET-SNN on image classification tasks from CIFAR and ImageNet datasets on VGG and ResNet architectures. We achieve top-1 accuracy of $69\%$ with 5 timesteps (inference latency) on the ImageNet dataset with $12\times$ less compute energy than an equivalent standard ANN. Additionally, DIET-SNN performs $20 - 500\times$ faster inference compared to other state-of-the-art SNN models.

## 1 Introduction

In recent years, a class of neural networks inspired by the event-driven form of computations in the brain has gained popularity for their promise of low-power computing (Painkras et al., 2013; Davies et al., 2018). Spiking neural networks (SNNs) first emerged in computational neuroscience as an attempt to model the behavior of biological neurons (Mainen & Sejnowski, 1995). They were pursued for low-complexity tasks implemented on bio-plausible neuromorphic platforms. At the same time in standard deep learning, the analog-valued artificial neural networks (ANNs) became the de-facto model for training various computer vision and natural language processing tasks (Krizhevsky et al., 2012; Hinton et al., 2012). The skyrocketing performance and success of multi-layer ANNs came at a significant power and energy cost (Li et al., 2016). Recently, major chip maker Nvidia estimated that $80 - 90\%$ of the energy cost of neural networks at data centers lies in inference processing (Freund, 2019). The tremendous energy costs and the demand for edge intelligence on battery-powered devices have shifted the focus on exploring lightweight energy-efficient inference models for machine intelligence. To that effect, various techniques such as weight pruning (Han et al., 2015), model compression (He et al., 2018), and quantization methods (Chakraborty et al., 2020) are proposed to reduce the size and computations in ANNs. Nonetheless, the inherent one-shot analog computation in ANNs requires the expensive operation of multiplying two real numbers (except when both weights and activations are 1-bit (Rastegari et al., 2016)). In contrast, SNNs inherently compute and

transmit information with binary signals distributed over time, providing a promising alternative for power-efficient machine intelligence.

For a long time, the success of SNNs was delayed due to the unavailability of good learning algorithms. But in recent years, the advent of supervised learning algorithms for SNN has overcome many of the roadblocks surrounding the discontinuous derivative of the spike activation function. Since SNNs receive and transmit information through spikes, analog values need to be encoded into spikes. There are a plethora of input encoding methods like rate coding (Diehl et al., 2015; Sengupta et al., 2019), temporal coding (Comsa et al., 2020), rank-order coding (Kheradpisheh & Masquelier, 2020), and other special coding schemes (Almomani et al., 2019). Among these, rate-coding has shown competitive performance on complex tasks (Diehl et al., 2015; Sengupta et al., 2019; Lee et al., 2019) while others are limited to simple tasks like learning the XOR function and classifying digits from the MNIST dataset. Also, dynamic vision sensors (DVS) record the change in image pixel intensities and directly convert it to spikes that can estimate optical flow (Lee et al., 2020) and classify hand gestures (Shrestha & Orchard, 2018). In rate coding, the analog value is represented by the rate of firing of the neuron. In each timestep, the neuron either fires (output '1') or stays inactive (output '0'). The number of timesteps[1] determines the discretization error in the representation of the analog value by spike-train. This leads to adopting a large number of timesteps for high accuracy at the expense of high inference latency (Sengupta et al., 2019). The two other parameters that are crucial for SNNs are firing *threshold* of the neuron and membrane potential *leak*. The neuron fires when the membrane potential exceeds the firing threshold and the potential is reset after each firing. Such neurons are usually referred to as integrate-and-fire (IF) neurons. The threshold value is very significant for the correct operation of SNNs because a high threshold will prevent the neuron from firing ('dead-neuron' problem), and a lower threshold will lead to unnecessary firing, affecting the ability of the neuron to differentiate between two input patterns. Another neuron model, leaky-integrate-and-fire (LIF), introduces a leak factor that allows the membrane potential to keep shrinking over time (Gerstner & Kistler, 2002). Most of the recent work on supervised learning in SNNs has either employed the IF or the LIF neuron model (Diehl et al., 2015; Sengupta et al., 2019; Lee et al., 2019; Rathi et al., 2020; Han et al., 2020). Some proposals adopt kernel-based spike response models (Huh & Sejnowski, 2018; Bohte et al., 2000), but for the most part, these approaches show limited performance on simple datasets and do not scale for deep networks. The leak provides an additional knob that can potentially be used to tune SNNs for better energy-efficiency. However, there has not been any exploration of the full design space of optimizing the leak and the threshold to achieve better latency (or energy) and accuracy tradeoff. Research, so far, has been mainly focused on using fixed leak for the entire network that can limit the capabilities of SNNs (Lee et al., 2019; Rathi et al., 2020). The firing thresholds are also fixed (Lee et al., 2019) or selected based on some heuristics (Sengupta et al., 2019; Rueckauer et al., 2017). Some recent works employ leak/threshold optimization but their application is limited to simple datasets (Fang, 2020; Yin et al., 2020). The current challenges in SNN models are high inference latency and energy, long training time, and high training costs in terms of memory and computation. Most of these challenges arise due to in-efficient input encoding, and improper methods of selecting the membrane leak and the threshold.

To address these challenges, this paper makes the following contributions:

- We propose a gradient descent based training method that learns the correct membrane leak and firing threshold for each layer of a deep spiking network via error-backpropagation. The goal is to jointly optimize the neuron parameters (membrane leak and threshold) and the network parameters (weights) to achieve high accuracy at low inference latency. The tailored membrane leak and threshold for each layer leads to large improvement in activation sparsity and energy-efficiency.

- We train the first convolutional layer to act as the spike-generator, whose spike-rate is a function of the weights, membrane leak, and threshold. This also eliminates the need for a generator function (and associated overheads) used in other coding schemes[2].

---

[1] Wall-clock time for 1 'timestep' is dependent on the number of computations performed and the underlying hardware (Frady et al., 2020). In simulation, 1 timestep is the time taken to perform 1 forward pass

[2] For rate-coding, a Poisson generator is used to convert the analog values to spike-train (Diehl et al., 2015). The encoder generates random numbers every timestep and compares it with the analog values to produce the spikes.

- To evaluate the effectiveness of the proposed algorithm, we train SNNs on both VGG (Simonyan & Zisserman, 2014) and ResNet (He et al., 2016) architectures for CIFAR (Krizhevsky et al., 2009) and ImageNet (Deng et al., 2009) datasets. DIET-SNN achieves similar accuracy as ANN with $6 - 18\times$ less compute energy. The performance is achieved at inference latency of 5 timesteps compared to $100 - 2000$ timesteps for state-of-the-art SNN models.

The source code implemented in Pytorch framework is available in the supplementary materials.

## 2 BACKGROUND AND RELATED WORK

The development of efficient learning algorithms for deep SNNs is an on-going research challenge. There has been a significant amount of success with recent supervised learning algorithms (Bellec et al., 2018; Wu et al., 2018; Sengupta et al., 2019; Wu et al., 2019b; Han et al., 2020) that can be broadly classified as conversion algorithms (Cao et al., 2015; Rueckauer et al., 2017; Sengupta et al., 2019) and spike-based backpropagation algorithms (Bellec et al., 2018; Lee et al., 2019). There are also some variants of bio-plausible algorithms that employ feedback alignment to update the weights. These algorithms apply random weights (Lillicrap et al., 2016) or fixed weights (Samadi et al., 2017) as the feedback weight during backpropagation[3]. The success of these algorithms is limited to simple tasks. Therefore, we focus our discussion on ANN-SNN conversion and backpropagation algorithms that are more suitable for complex tasks and are scalable to deep networks.

**ANN-SNN Conversion:** ANN-SNN conversion is the most successful method of training rate-coded deep SNNs (Cao et al., 2015; Diehl et al., 2015; Rueckauer et al., 2017; Sengupta et al., 2019; Han et al., 2020). In this process, an ANN with ReLU neurons is trained with standard backpropagation with some restrictions (no bias, average pooling, no batch normalization), although some works have shown that some of the restrictions can be relaxed (Rueckauer et al., 2017). Next, an SNN with IF neurons and iso-architecture as ANN is initialized with the weights of the trained ANN. The underlying principle of this process is that a ReLU neuron can be mapped to an IF neuron with minimum loss. The mapping is possible for SNNs that operate on rate-coded inputs (Diehl et al., 2015; Sengupta et al., 2019) or on direct input encoding (Rueckauer et al., 2017). The major bottleneck of this method is to determine the firing threshold of the IF neurons that can balance the accuracy-latency tradeoff. Generally, the threshold is computed as the maximum pre-activation of the IF neuron resulting in high inference accuracy at the cost of high inference latency ($2000 - 2500$ timesteps) (Sengupta et al., 2019). In recent work, the authors showed that instead of using the maximum pre-activation, a certain percentile of the pre-activation distribution reduces the inference latency ($100 - 200$ timesteps) with minimal accuracy drop (Lu & Sengupta, 2020). These heuristic techniques of determining the firing threshold lead to a sub-optimal accuracy-latency tradeoff. Additionally, ANN-SNN conversion has a major drawback: the absence of the timing information. The quintessential parameter 'time' is not utilized in the conversion process which leads to higher inference latency.

**Error Backpropagation in SNN:** ANNs have achieved success with gradient-based training that backpropagates the error signal from the output layer to the input layer. It requires computing a gradient of each operation performed in the forward pass. Unfortunately, the IF and the LIF neuron does not have a continuous derivative. The derivative of the spike function (Dirac delta) is undefined at the time of spike and '0' otherwise. This has hindered the application of standard backpropagation in SNN. There have been many proposals to perform gradient-based training in SNNs (Huh & Sejnowski, 2018; Lee et al., 2019) – among them, the most successful is surrogate-gradient based optimization (Neftci et al., 2019). The discontinuous derivative of the IF neuron is approximated by a continuous function that serves as the surrogate for the real gradient. SNNs trained with surrogate-gradient perform backpropagation through time (BPTT) to achieve high accuracy and low latency (100 timesteps), but the training is very compute and memory intensive in terms of total training iterations compared to conversion techniques. The multiple-iteration training effort with exploding memory requirement for backpropagation has limited the application of this method to simpler tasks on shallow architectures (Lee et al., 2019).

---

[3]In standard backpropagation, the feedback weight is $\boldsymbol{W}^T$, where $\boldsymbol{W}$ is the weight used in the forward pass

**Hybrid SNN Training:** In recent work, the authors proposed a hybrid mechanism to circumvent the high training costs of backpropagation as well as maintain low inference latency ($100-250$ timesteps) (Rathi et al., 2020). The method involves both ANN-SNN conversion and error-backpropagation. A trained ANN is converted to an SNN as described earlier and the weights of the converted SNN are further fine-tuned with surrogate gradient and BPTT. The authors showed a faster convergence ($< 20$ epochs) in SNN training due to the precursory initialization from the ANN-SNN conversion. This presents a practically feasible method to train deep SNNs with limited resources, which is otherwise challenging with only backpropagation from random initialization (Lee et al., 2019). Hybrid training tries to achieve the best of both worlds: high accuracy and low latency. But it still employs rate coding, fixed membrane leak, and fixed threshold, and therefore, the latency-accuracy tradeoff can be improved further.

In this work, we adopt the hybrid training method to train the SNNs. We start with ANN-SNN conversion and select the threshold for each layer as 95 percentile of the pre-activation distribution. The pixel intensities are directly applied in the input layer during the threshold computation. This serves as the initial model that is further trained to optimize the membrane leak and threshold.

## 3 ALGORITHM FOR TRAINING DIET-SNN

**Direct Input Encoding:** The pixel intensities of an image are applied directly to the input layer of the SNN at each timestep (Rueckauer et al., 2017; Lu & Sengupta, 2020). The first convolutional layer composed of LIF neurons acts as both the feature extractor and the spike-generator, which accumulates the weighted pixel values and generates output spikes. This is similar to rate-coding, but the spike-rate is a function of the weights, membrane leak, and threshold that are all learned by gradient-descent.

**Neuron Model:** We employ the LIF neuron model described by

$$u_i^t = \lambda_i u_i^{t-1} + \sum_j w_{ij} o_j^t - v_i o_i^{t-1} \tag{1}$$

$$z_i^{t-1} = \frac{u_i^{t-1}}{v_i} - 1 \quad \text{and} \quad o_i^{t-1} = \begin{cases} 1, & \text{if } z_i^{t-1} > 0 \\ 0, & \text{otherwise} \end{cases} \tag{2}$$

where $u$ is the membrane potential, $\lambda$ is the leak factor with a value in $[0-1]$, $w$ is the weight connecting pre-neuron $j$ and post-neuron $i$, $o$ is the binary spike output, $v$ is the firing threshold, and $t$ represents the timestep. The first term in Equation 1 denotes the leakage in the membrane potential, the second term integrates the weighted input received from pre-neuron, and the third term accounts for the reduction in potential when the neuron generates an output spike. After the spike, a soft reset is performed where the potential is reduced by threshold instead of resetting to zero (Han et al., 2020). The threshold governs the average integration time of input, and the leak regulates how much of the potential is retained from the previous timestep. Now, we derive the expressions to compute the gradients of the parameters at all layers. The spatial and temporal credit assignment is performed by unrolling the network in time and employing BPTT.

**Output layer:** The neuron model in the output layer only accumulates the incoming inputs without any leakage and does not generate an output spike and is described by

$$\boldsymbol{u}_l^t = \boldsymbol{u}_l^{t-1} + \boldsymbol{W}_l \boldsymbol{o}_{l-1}^t \tag{3}$$

where $\boldsymbol{u}_l$ is a vector containing the membrane potential of $N$ output neurons, $N$ is the number of classes in the task, $\boldsymbol{W}_l$ is the weight matrix connecting the output layer and the previous layer, and $\boldsymbol{o}_{l-1}$ is a vector containing the spike signals from layer $(l-1)$. The loss function is defined on $\boldsymbol{u}_l$ at the last timestep $T$. We employ the cross-entropy loss and the softmax is computed on $\boldsymbol{u}_l^T$. The symbol $T$ is used for timestep and not to denote the transpose of a matrix.

$$\boldsymbol{s}(\boldsymbol{u}_l^T) : \begin{bmatrix} u_1^T \\ \dots \\ u_N^T \end{bmatrix} \rightarrow \begin{bmatrix} s_1 \\ \dots \\ s_N \end{bmatrix} \quad s_i = \frac{e^{u_i^T}}{\sum_{k=1}^N e^{u_k^T}} \tag{4}$$

Table 1: Top-1 classification accuracy

| Architecture | ANN | ANN-SNN | Weight Optimization | **DIET-SNN** | Timesteps ($T$) |
|---|---|---|---|---|---|
| | | | CIFAR10 | | |
| VGG6 | 90.80% | 86.19% | 89.24% | **89.42%** | 5 |
| | | | | **90.05%** | 10 |
| VGG16 | 93.72% | 73.52% | 91.68% | **92.70%** | 5 |
| | | | | **93.44%** | 10 |
| ResNet20 | 92.79% | 47.26% | 90.29% | **91.78%** | 5 |
| | | | | **92.54%** | 10 |
| | | | CIFAR100 | | |
| VGG16 | 71.82% | 46.54% | 65.83% | **69.67%** | 5 |
| ResNet20 | 64.64% | 31.40% | 62.95% | **64.07%** | 5 |
| | | | ImageNet | | |
| VGG16 | 70.08% | 24.58% | 64.32% | **69.00%** | 5 |

$$L = -\sum_i y_i log(s_i), \quad \frac{\partial L}{\partial \boldsymbol{u}_l^T} = \boldsymbol{s} - \boldsymbol{y} \tag{5}$$

where $\boldsymbol{s}$ is the vector containing the softmax values, $L$ is the loss function, and $\boldsymbol{y}$ is the one-hot encoded vector of the true label or target. The weight update is computed as

$$\boldsymbol{W}_l = \boldsymbol{W}_l - \eta \Delta \boldsymbol{W}_l \tag{6}$$

$$\Delta \boldsymbol{W}_l = \sum_t \frac{\partial L}{\partial \boldsymbol{W}_l} = \sum_t \frac{\partial L}{\partial \boldsymbol{u}_l^t} \frac{\partial \boldsymbol{u}_l^t}{\partial \boldsymbol{W}_l} = \frac{\partial L}{\partial \boldsymbol{u}_l^T} \sum_t \frac{\partial \boldsymbol{u}_l^t}{\partial \boldsymbol{W}_l} = (\boldsymbol{s} - \boldsymbol{y}) \sum_t \boldsymbol{o}_{l-1}^t \tag{7}$$

$$\frac{\partial L}{\partial \boldsymbol{o}_{l-1}^t} = \frac{\partial L}{\partial \boldsymbol{u}_l^T} \frac{\partial \boldsymbol{u}_l^t}{\partial \boldsymbol{o}_{l-1}^t} = (\boldsymbol{s} - \boldsymbol{y}) \boldsymbol{W}_l \tag{8}$$

where $\eta$ is the learning rate.

**Hidden layers:** The neurons in the convolutional and fully-connected layers are defined by the LIF model as

$$\boldsymbol{u}_l^t = \lambda_l \boldsymbol{u}_l^{t-1} + \boldsymbol{W}_l \boldsymbol{o}_{l-1}^t - v_l \boldsymbol{o}_l^{t-1} \tag{9}$$

$$\boldsymbol{z}_l^t = \frac{\boldsymbol{u}_l^t}{v_l} - 1 \quad \text{and} \quad \boldsymbol{o}_l^t = \begin{cases} 1, & \text{if } \boldsymbol{z}_l^t > 0 \\ 0, & \text{otherwise} \end{cases} \tag{10}$$

where $\lambda_l$ ($v_l$) is a real value representing leak (threshold) for all neurons in layer $l$. All neurons in a layer share the same leak and threshold value. This reduces the number of trainable parameters and we did not observe any significant improvement by assigning individual threshold/leak to each neuron. The weight update is calculated as

$$\Delta \boldsymbol{W}_l = \sum_t \frac{\partial L}{\partial \boldsymbol{W}_l} = \sum_t \frac{\partial L}{\partial \boldsymbol{o}_l^t} \frac{\partial \boldsymbol{o}_l^t}{\partial \boldsymbol{z}_l^t} \frac{\partial \boldsymbol{z}_l^t}{\partial \boldsymbol{u}_l^t} \frac{\partial \boldsymbol{u}_l^t}{\partial \boldsymbol{W}_l} = \sum_t \frac{\partial L}{\partial \boldsymbol{o}_l^t} \frac{\partial \boldsymbol{o}_l^t}{\partial \boldsymbol{z}_l^t} \frac{1}{v_l} \boldsymbol{o}_{l-1}^t \tag{11}$$

$\partial \boldsymbol{o}_l^t / \partial \boldsymbol{z}_l^t$ is the discontinuous gradient and we approximate it with the surrogate gradient (Bellec et al., 2018)

$$\frac{\partial \boldsymbol{o}_l^t}{\partial \boldsymbol{z}_l^t} = \gamma \, max\{0, 1 - |\boldsymbol{z}_l^t|\} \quad \text{and} \quad \frac{\partial \boldsymbol{o}_l^t}{\partial \boldsymbol{u}_l^t} = \frac{\partial \boldsymbol{o}_l^t}{\partial \boldsymbol{z}_l^t} \frac{\partial \boldsymbol{z}_l^t}{\partial \boldsymbol{u}_l^t} = \frac{\partial \boldsymbol{o}_l^t}{\partial \boldsymbol{z}_l^t} \frac{1}{v_l} \tag{12}$$

where $\gamma$ is a constant denoting the maximum value of the gradient. The threshold update is then computed as

$$v_l = v_l - \eta \Delta v_l \tag{13}$$

$$\Delta v_l = \sum_t \frac{\partial L}{\partial v_l} = \sum_t \frac{\partial L}{\partial \boldsymbol{o}_l^t} \frac{\partial \boldsymbol{o}_l^t}{\partial \boldsymbol{z}_l^t} \frac{\partial \boldsymbol{z}_l^t}{\partial v_l} = \sum_t \frac{\partial L}{\partial \boldsymbol{o}_l^t} \frac{\partial \boldsymbol{o}_l^t}{\partial \boldsymbol{z}_l^t} \left( \frac{-v_l \boldsymbol{o}_l^{t-1} - \boldsymbol{u}_l^t}{(v_l)^2} \right) \tag{14}$$

And finally the leak update is computed as

$$\lambda_l = \lambda_l - \eta \Delta \lambda_l \quad \text{and} \quad \Delta \lambda_l = \sum_t \frac{\partial L}{\partial \lambda_l} = \sum_t \frac{\partial L}{\partial \boldsymbol{o}_l^t} \frac{\partial \boldsymbol{o}_l^t}{\partial \boldsymbol{u}_l^t} \frac{\partial \boldsymbol{u}_l^t}{\partial \lambda_l} = \sum_t \frac{\partial L}{\partial \boldsymbol{o}_l^t} \frac{\partial \boldsymbol{o}_l^t}{\partial \boldsymbol{u}_l^t} \boldsymbol{u}_l^{t-1} \tag{15}$$

## 4 EXPERIMENTS

Table 2: DIET-SNN compared with other SNN models

| Model | Method | Architecture | SNN Accuracy | Timesteps |
|-------|--------|--------------|--------------|-----------|
| CIFAR10 | | | | |
| Sengupta et al. (2019) | ANN-SNN | VGG16 | 91.55% | 2500 |
| Rueckauer et al. (2017) | ANN-SNN | 4 Conv, 2 FC | 90.85% | 400 |
| Rathi et al. (2020) | Hybrid | VGG16 | 92.02% | 200 |
| Lee et al. (2019) | Backprop | VGG9 | 90.45% | 100 |
| Wu et al. (2019b) | Backprop | CIFARNet | 90.53% | 12 |
| Wu et al. (2019a) | Backprop | CIFARNet | 90.98% | 8 |
| Zhang & Li (2020) | Backprop | CIFARNet | 91.41% | 5 |
| **This work** | **DIET-SNN** | **CIFARNet** | **91.59%** | **5** |
| **This work** | **DIET-SNN** | **VGG16** | **92.70%** | **5** |
| CIFAR100 | | | | |
| Han et al. (2020) | ANN-SNN | VGG16 | 70.09% | 768 |
| Rathi et al. (2020) | Hybrid | VGG11 | 67.87% | 125 |
| Lu & Sengupta (2020) | ANN-SNN | VGG15 | 63.20% | 62 |
| **This work** | **DIET-SNN** | **VGG16** | **69.67%** | **5** |
| ImageNet | | | | |
| Sengupta et al. (2019) | ANN-SNN | VGG16 | 69.96% | 2500 |
| Han et al. (2020) | ANN-SNN | VGG16 | 71.34% | 768 |
| Rueckauer et al. (2017) | ANN-SNN | VGG16 | 49.61% | 400 |
| Rathi et al. (2020) | Hybrid | VGG16 | 65.19% | 250 |
| Lu & Sengupta (2020) | ANN-SNN | VGG15 | 66.56% | 64 |
| Wu et al. (2019a) | Backprop | AlexNet | 50.22% | 10 |
| **This work** | **DIET-SNN** | **VGG16** | **69.00%** | **5** |

The three-step DIET-SNN training pipeline (Fig. 1) begins with training an ANN without the bias term and batch-normalization to achieve minimal loss during ANN-SNN conversion (Diehl et al., 2015; Sengupta et al., 2019; Rathi et al., 2020). Dropout (Srivastava et al., 2014) is used as the regularizer and the dropout mask is unchanged during all timesteps of an input sample (Rathi et al., 2020). Average-pooling is used to reduce the feature map size in VGG architectures, whereas for ResNets, a stride of 2 is employed to reduce the feature size. Supplementary material contains the complete architecture details, training hyperparameters, and dataset descriptions. Next, the trained ANN is converted to SNN with IF neurons. The threshold for each layer is computed sequentially as the 99.7 percentile

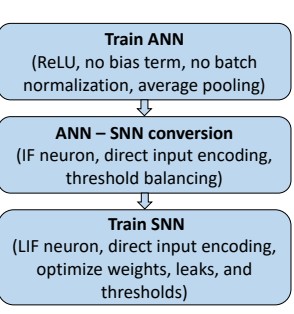

Figure 1: Training pipeline

of the pre-activation distribution at each layer. The pre-activation for each neuron is the weighted sum of inputs $\sum_j o_j w_{ij}$ received by the neuron. During threshold computation, the leak in the hidden layers is set to unity, and the input layer employs direct input encoding. Finally, the converted SNN is trained with error-backpropagation to optimize the weights, the membrane leak, and the firing thresholds of each layer as described by the equations in Section 3. We evaluated the performance of these networks on VGG and ResNet architectures for CIFAR and ImageNet datasets. Column-2 in Table 1 shows the ANN accuracy; column-3 shows the accuracy after ANN-SNN conversion with $T$ timesteps; column-4 shows the accuracy when only the weights in SNN are trained with spike-based backpropagation; column-5 shows the accuracy when the weights, threshold, and leak are jointly optimized (DIET-SNN). The performance of DIET-SNN compared to current state-of-the-art SNNs is shown in Table 2. DIET-SNN shows $5 - 100\times$ improvement in inference latency compared to other spiking networks. The authors in Wu et al. (2019b) achieved convergence in 12 timesteps with a special input encoding layer, but the extra overhead may affect the overall latency and energy.

## 5 ENERGY EFFICIENCY

In ANN, each operation computes a dot-product involving one floating-point (FP) multiplication and one FP addition (MAC), whereas, in SNN, each operation is only one FP addition due to binary spikes. The computations in SNN implemented on neuromorphic hardware are event-driven (Davies et al., 2018; Frady et al., 2020). Therefore, in the absence of spikes, there are no computations and no

Table 3: ANN vs DIET-SNN compute energy. Each operation in ANN (SNN) consumes $4.6pJ$ $(0.9pJ)$. The input layer in DIET-SNN is non-spiking, so it's energy is same as ANN. Column-5 shows the ratio of #operations in input layer to the total #operations in the network.

| Architecture (timesteps) | Dataset | Normalized $\#OP_{ANN}(a)$ | Normalized $\#OP_{SNN}(b)$ | $\frac{\#OP \text{ layer } 1}{\text{Total } \#OP}(c)$ | ANN / DIET-SNN Energy $\left(\frac{a*4.6}{c*4.6+(1-c)*b*0.9}\right)$ |
|---|---|---|---|---|---|
| VGG6 (T=5) | CIFAR10 | 1.0 | 0.14 | 0.029 | 18 |
| VGG16 (T=5) | CIFAR10 | 1.0 | 0.39 | 0.005 | 12.4 |
| VGG16 (T=5) | CIFAR100 | 1.0 | 0.40 | 0.005 | 12.1 |
| VGG16 (T=5) | ImageNet | 1.0 | 0.41 | 0.006 | 11.7 |
| ResNet20 (T=5) | CIFAR10 | 1.0 | 0.76 | 0.013 | 6.3 |
| ResNet20 (T=5) | CIFAR100 | 1.0 | 0.72 | 0.013 | 6.6 |

active energy is consumed. We computed the energy cost/operation for ANNs and SNNs in 45nm CMOS technology. The energy cost for 32-bit ANN MAC operation $(4.6pJ)$ is $5.1\times$ more than SNN addition operation $(0.9pJ)$ (Horowitz, 2014). These numbers may vary for different technologies, but generally, in most technologies, the addition operation is much cheaper than the multiplication operation. In ANN, the number of operations/layer is defined by

$$\#OP_{ANN} = \begin{cases} k_w \times k_h \times c_{in} \times h_{out} \times w_{out} \times c_{out}, & \text{Convolution layer} \\ f_{in} \times f_{out}, & \text{Fully-connected layer} \end{cases} \tag{16}$$

where $k_w(k_h)$ is kernel width (height), $c_{in}(c_{out})$ is the number of input (output) channels, $h_{out}(w_{out})$ is the height (width) of the output feature map, and $f_{in}(f_{out})$ is the number of input (output) features. The number of operations/layer in iso-architecture SNN is specified by

$$\#OP_{SNN} = SpikeRate_l \times \#OP_{ANN} \tag{17}$$

$$SpikeRate_l = \frac{\#TotalSpikes_l \text{ over all inference timesteps}}{\#Neurons_l} \tag{18}$$

where $SpikeRate_l$ is the total spikes in layer $l$ over all timesteps averaged over the number of neurons in layer $l$. A spike rate of 1 (every neuron fired once) implies that the number of operations for ANN and SNN are the same (though operations are MAC in ANN while addition in SNNs). Lower spike rates denote more sparsity in spike events and higher energy-efficiency. The average spike rate for VGG16 during inference is around 1.6 (Fig. 2(a)), indicating that DIET-SNN is more energy-efficient than ANN. In deeper layers, the leak decreases while the threshold increases (Fig. 2(b)), resulting in lower spike rates and better energy-efficiency.

Table 3 shows the compute energy comparison of ANN and DIET-SNN. The energy for ResNet is more than VGG because more than $50\%$ of the total operations in ResNet occurs in the first 3 layers where the spike rate is high. The layerwise spike rate in ResNet is provided in the supplementary section. The standard ResNet architecture was modified with initial 3 plain convolutional layers to minimize the accuracy loss during ANN-SNN conversion (Sengupta et al., 2019). In Lu & Sengupta (2020) an average spike rate of 2.35 for VGG16 on CIFAR100 was reported with $62\%$ accuracy. The maximum spike rate of 20 was reported for VGG16 architecture on CIFAR10 dataset (Rathi et al., 2020). DIET-SNN performs considerably better in all metrics compared to these models and achieves better compute energy than ANN on complex tasks like CIFAR and ImageNet with similar accuracy. We did not consider the data movement cost in our evaluation as it is dependent on the system architecture and the underlying hardware implementation. Although we would like to mention that in SNN the membrane potentials have to be fetched at every timestep, in addition to the weights and activations. Many proposals reduce the memory cost by data buffering (Shen et al., 2017), trading computations for memory (Chen et al., 2016a), and data reuse through efficient dataflows (Chen et al., 2016b). All such techniques can be extended to SNNs to address the memory cost. The training of SNNs is still a cause of concern for energy-efficiency because it requires several days, even on high-performance GPUs. The hybrid approach and DIET-SNN alleviate the issue by reducing the number of training epochs and the number of timesteps, but further innovations in both algorithms and accelerators for SNNs are required to reduce the training cost.

## 6 EFFECT OF DIRECT INPUT ENCODING AND THRESHOLD/LEAK OPTIMIZATION

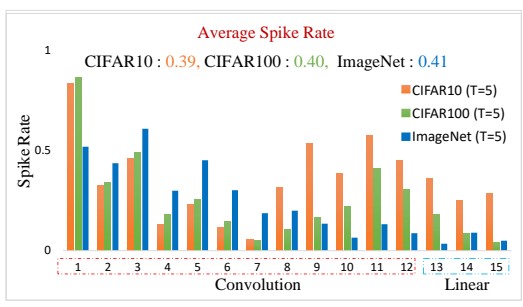

(a) Spike Rate

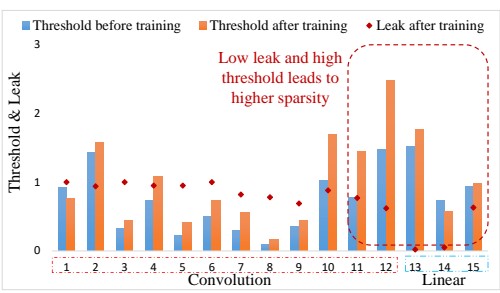

(b) Leak and Threshold after training

Figure 2: (a) Layerwise spike rate for VGG16 during inference over entire test-set. Average spike rate is calculated as total $\#spikes/\#neurons$. An average spike rate of $1.65$ indicates that every neuron fired on average $1.65$ times for each image over all timesteps. (b) Layerwise leak and threshold for VGG16 on CIFAR100 dataset. The threshold before training represents the values obtained from ANN-SNN conversion process. The leak before training is unity for all layers.

Table 1 shows the effect on accuracy (under iso-timesteps) by training threshold/leak along with direct input encoding. In this section, we analyze the impact of input encoding, threshold and leak on the average spike rate and latency (under iso-accuracy). We train four different spiking networks: (a) SNN with IF neuron and Poisson rate encoding; (b) SNN with IF neuron and direct input encoding; (c) threshold optimization added to (b); (d) SNN with LIF neuron, analog encoding, and threshold/leak optimization (DIET-SNN). The SNNs are trained to achieve similar accuracy for VGG16 on CIFAR10.

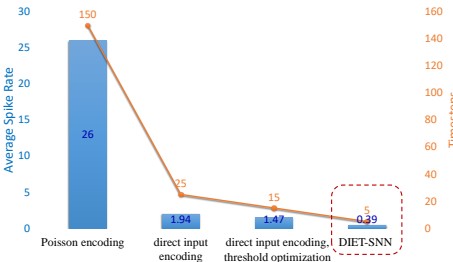

Figure 3: Effect of employing direct input encoding, threshold and leak optimization

The networks (a)-(d) achieved an accuracy of $92.10\%$, $92.41\%$, $92.37\%$, and $92.70\%$, respectively. The network with Poisson input encoding required $150$ timesteps with average spike rate of $26$ (Fig. 3). By replacing Poisson encoding with direct input encoding (proposed work), the latency (spike-rate) improved to $25$ timesteps ($1.94$). As mentioned in Section 1, the information in rate-coded SNNs is encoded in the firing rate of the neuron. In Poisson encoding, the firing rate is proportional to the pixel value, whereas in direct input encoding, the SNN learns the optimal firing rate by training the parameters of the first convolutional layer. This reduces the number of timesteps required to encode the input. Next, the addition of threshold optimization reduces the latency (spike-rate) to $15$ timesteps ($1.47$). In SNNs with IF neurons, a neuron's activity is dependent on the ratio of the weights and the neuron's threshold. Therefore, training only weights should be sufficient, however, training threshold along with weights provides additional parameter for optimization and leads to lower latency as shown in network (c) compared to (b). Finally, the addition of the leak parameter reduces the latency (spike-rate) to $5$ timesteps ($0.39$). The leak and threshold together eliminate the excess membrane potential and suppress unnecessary firing activities that improve the latency and the spike-rate. The compute energy compared to ANN (Table 3) for the networks (a)-(d) are $0.2$, $2.6$, $3.4$, and $12.4$, respectively. Therefore, the co-optimization of weights, leak, and threshold along with direct input encoding provide an SNN with improved latency and low energy-consumption.

## 7 CONCLUSIONS

SNNs that operate with asynchronous discrete events can potentially solve the energy issue in deep learning. To that effect, we presented DIET-SNN, an energy-efficient spiking network that is trained to operate with low inference latency and high activation sparsity. The membrane leak and the firing threshold of the LIF neurons are trained with error-backpropagation along with the weights of the network to optimize both accuracy and latency. We initialize the parameters of DIET-SNN, taken from a trained ANN, to speed-up the training with spike-based backpropagation. The image pixels are applied directly as input to the network, and the first convolutional layer is trained to perform the spike-generation operation. This leads to high activation sparsity in the convolutional and dense layers of the network. The high sparsity combined with low inference latency reduces the compute energy by $2-8\times$ compared to an equivalent ANN with similar accuracy. DIET-SNN achieves similar accuracy as other state-of-the-art SNN models with $5-100\times$ less number of timesteps.

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
