# OpenReview forum: "DIET-SNN: A Low-Latency Spiking Neural Network with Direct Input Encoding & Leakage and Threshold Optimization"
_ICLR.cc/2021/Conference — Reject_

### Official Review · AnonReviewer2 · 2020-10-19
**Training SNNs for visual categorization with few timesteps**

**Rating:** 6
**Confidence:** 4

**Review:**

Tile: training SNNs for visual categorization with few timesteps

PROS
* high accuracy with only 20-25 timesteps

CONS
* nothing really new

The authors train convolutional SNNs for image classification, using surrogate gradient learning (SGL). They combine 4 mechanisms:

1) Hybrid SNN Training. First train an ANN with the same architecture, and use the resulting weights as initial values for SGL, to accelerate convergence. This has already been proposed in (Rathi et al., 2020).
2) Direct Input Encoding. Instead of converting the image RGB values into spike trains using Poisson rate coding, the authors feed the analog RGB values in the first convolutional layer, which treats them as input current, and emits spikes using the LIF neuron model. This allows using fewer time steps, since Poisson rate coding imposes long time windows to estimate rates and average out noise. But again, this is not new. It's been done for example in (Rueckauer et al., 2017; Lu & Sengupta, 2020), which they cite.
3) Leak timescale training. The time scale of the leak is an important parameter in SNNs. It can be trained by SGL, just like the weights. Again this is not new. See for example (which should be cited):
Fang W (2020) Leaky Integrate-and-Fire Spiking Neuron with Learnable Membrane Time Parameter.
Yin B, Corradi F, Bohté SM (2020) Effective and Efficient Computation with Multiple-timescale Spiking Recurrent Neural Networks. arXiv.
Zimmer R, Pellegrini T, Singh Fateh S, Masquelier T (2019) Technical report: supervised training of convolutional spiking neural networks with PyTorch. arXiv.
4) Threshold training. Again this is not new. For example Zimmer et al 2019 (ref above) did it already. Also, training both the weights and the thresholds seems redundant, since only the ratio between weights and threshold matters. About that the author say:
"Intuitively, the effect of optimizing either weights or thresholds in SNNs with
IF neurons should have similar consequences. But the threshold affects the activity of each neuron, whereas the weights are shared among multiple neurons."
I don't understand this sentence. Weights are shared only in the convolutional layers. Plus earlier the authors said that the threshold is the same for all neurons of a given layer.

So in short, the authors successfully combine known approaches, but do not propose anything new at a conceptual or theoretical level. In my opinion this paper is mostly an engineering effort. That being said, it seems that no one can claim a better accuracy on CIFAR and ImageNet using this nb of timesteps (or fewer).

MINOR POINTS:

* How much longer is convergence if the authors skip the ANN training, and start the SGL training from random weights?
* p2: "The length of timesteps" -> "The number of timesteps"
* "The neurons in the convolutional and linear layers are defined by the LIF model" I guess the authors meant "dense" or "fully connected" instead of "linear", since the LIF is not linear
* I liked a lot the "iso-accuracy" analysis of section 6. But I suggest the authors do the same for their energy analysis of section 5. The SNNs consume less than the ANNs (Table 3) but are also less accurate (Table 1).

---

> ### Author Response · Authors · 2020-11-19
> **Response to Reviewer #2 Part 1/1**
>
> **Reviewer's comment:** Leak timescale training. The time scale of the leak is an important parameter in SNNs. It can be trained by SGL, just like the weights. Again this is not new. See for example (which should be cited): Fang W (2020) Leaky Integrate-and-Fire Spiking Neuron with Learnable Membrane Time Parameter. Yin B, Corradi F, Bohté SM (2020) Effective and Efficient Computation with Multiple-timescale Spiking Recurrent Neural Networks. arXiv. Zimmer R, Pellegrini T, Singh Fateh S, Masquelier T (2019) Technical report: supervised training of convolutional spiking neural networks with PyTorch. ArXiv.
>
> **Author's response:** Thank you for pointing this out. We have cited the mentioned works in the updated manuscript. Although the idea of training leak has been proposed elsewhere, its application on deeper networks and challenging datasets have not been shown before. Though Fang W et al. achieve competitive accuracy on the CIFAR10 dataset, they fail to mention the number of timesteps used in their experiments. Yin B et al. and Zimmer R et al. do not show results for any of the three datasets (CIFAR10, CIFAR100, ImageNet) considered in our work.
>
> **Reviewer's comment:** Threshold training. Again this is not new. For example Zimmer et al 2019 (ref above) did it already. Also, training both the weights and the thresholds seems redundant, since only the ratio between weights and threshold matters. About that the author say: "Intuitively, the effect of optimizing either weights or thresholds in SNNs with IF neurons should have similar consequences. But the threshold affects the activity of each neuron, whereas the weights are shared among multiple neurons." I don't understand this sentence. Weights are shared only in the convolutional layers. Plus earlier the authors said that the threshold is the same for all neurons of a given layer.
>
> **Author's response:** Yes, we were referring to convolutional layers (updated the writing in manuscript). The activity of a neuron can be modulated by changing its threshold. To achieve the same effect by modulating weights, multiple weights need to be changed both for fully-connected or convolutional layers. Additionally in convolutional layers, as the weights are shared, it may affect the activity of other neurons as well. You are right, this argument may not hold if we have the same threshold for all neurons in a layer. But we did notice a lower number of timesteps in the network where both the weights and threshold were trained jointly (network (b) and (c) in Section 6). The reason may be that the optimizer is able to find a better setting when both the parameters are tunable. We did try having an individual threshold for each neuron but that did not lead to better performance. We continue to investigate this with different optimizers.
>
> **Reviewer's comment:** So in short, the authors successfully combine known approaches, but do not propose anything new at a conceptual or theoretical level. In my opinion this paper is mostly an engineering effort. That being said, it seems that no one can claim a better accuracy on CIFAR and ImageNet using this nb of timesteps (or fewer).
>
> **Author's response:** We agree some of the concepts are proposed in other works as well, but as you mentioned, in our work we implement the methods on large-scale datasets and thereby addressing the engineering challenges which were not covered in previous works. This work achieves the best performance on CIFAR and ImageNet datasets with lower number of timesteps.
>
> **MINOR POINTS:**
> * How much longer is convergence if the authors skip the ANN training, and start the SGL training from random weights?
>
> We did train a network with 4 convolutional layers and 2 fully-connected layers from scratch and compared its performance with hybrid learning. The network from scratch required more than 300 epochs of training whereas the hybrid method needed only 100 epochs of training with spike-based backpropagation.
>
> * p2: "The length of timesteps" -> "The number of timesteps"
>
> Thank you for pointing this out, we have updated it in the manuscript.
>
> * "The neurons in the convolutional and linear layers are defined by the LIF model" I guess the authors meant "dense" or "fully connected" instead of "linear", since the LIF is not linear
>
> Yes, you are correct. We have updated the manuscript and refer it as fully-connected layers.
>
> * I liked a lot the "iso-accuracy" analysis of section 6. But I suggest the authors do the same for their energy analysis of section 5. The SNNs consume less than the ANNs (Table 3) but are also less accurate (Table 1).
>
> The ANN/SNN energy ratio for the networks (a)-(d) in Section 6 are 0.2, 2.6, 3.4, and 12.4, respectively. We have updated Section 6 in the manuscript with the energy numbers. Yes, we agree that ANNs generally achieve higher accuracy. We have updated the results in Table-1 where the gap between ANN and SNN accuracies is reduced.

---

> > ### Comment · AnonReviewer2 · 2020-11-20
> > **The ms has been improved.**
> >
> > The authors have taken my main comments into account, and I think the new version of the ms has been improved. My overall impression remains the same: the paper does not propose any conceptual advance, but combines and scales up known mechanisms, leading to an impressive accuracy on challenging datasets like ImageNet.

---

> > > ### Author Response · Authors · 2020-11-20
> > > **Thank You**
> > >
> > > Thank you for replying to our comments. We appreciate your time and effort to help us improve the manuscript.

---

### Official Review · AnonReviewer4 · 2020-10-23
**This paper proposed a hybrid of ANN-SNN conversion and direct SNN training. It can train the network with small number of time steps.**

**Rating:** 3
**Confidence:** 4

**Review:**

Strength:
I appreciate the experimental results demonstrated on challenging datasets like CIFAR100 and ImageNet.

Weakness
(1) There are two existing papers emphasizing direct training SNN with extremely low latency [1][2]. Therefore, the authors should comment more on how the proposed method is different or is better compared to these two papers (they use a smaller number of time steps.). In addition, I hope the authors to show the performance comparison with these two references. In the experimental results, the paper compares performance with [1]. However, the network size is significantly different. I think a comparison of the same network size can help to demonstrate the effectiveness of the proposed method.

(2) To my understanding, the method proposed in this paper has nothing different compared to the existing BPTT method with the surrogate gradient. I hope the authors can claim clearly the novelty of this paper. For example, how is the proposed method different or better from [3] and [4]. As far as I know, the only difference is that the threshold and leaky parameters are also trained in this paper. However, this can also be easily done in other existing methods like [2][3][4]. There's no doubt their performance can also be improved since it introduces more tunable parameters.

(3) Tuning threshold and leaky parameters may be unbiologically plausible.


(4) What is the weight optimization method? As shown in Table 1, it seems the weight optimization contributes more to the performance than the proposed method.


[1] Wu, Y., Deng, L., Li, G., Zhu, J., Xie, Y., & Shi, L. (2019, July). Direct training for spiking neural networks: Faster, larger, better. In Proceedings of the AAAI Conference on Artificial Intelligence (Vol. 33, pp. 1311-1318).
[2] Zhang, W., & Li, P. (2020). Temporal Spike Sequence Learning via Backpropagation for Deep Spiking Neural Networks. arXiv preprint arXiv:2002.10085.
[3] Wu, Y., Deng, L., Li, G., Zhu, J., & Shi, L. (2018). Spatio-temporal backpropagation for training high-performance spiking neural networks. Frontiers in neuroscience, 12, 331.
[4] Shrestha, S. B., & Orchard, G. (2018). Slayer: Spike layer error reassignment in time. In Advances in Neural Information Processing Systems (pp. 1412-1421).

---

> ### Author Response · Authors · 2020-11-19
> **Response to Reviewer #4 Part 1/2**
>
> **Reviewer's comment:** Weakness (1) There are two existing papers emphasizing direct training SNN with extremely low latency [1][2]. Therefore, the authors should comment more on how the proposed method is different or is better compared to these two papers (they use a smaller number of time steps.). In addition, I hope the authors to show the performance comparison with these two references. In the experimental results, the paper compares performance with [1]. However, the network size is significantly different. I think a comparison of the same network size can help to demonstrate the effectiveness of the proposed method.
>
> **Author's response:**  The authors in [1] proposed a normalization method (NeuNorm) that computes a weighted summation of spike count and uses that quantity as the input to the convolutional layer instead of the raw spike signals. Therefore, the convolution requires the multiply-and-accumulate (MAC) operation as both the input and the weight are real-valued quantities. In SNN, the major advantage is that the expensive MAC operation (needed in ANN) is reduced to simple additions due to binary inputs. Although the authors achieved competitive accuracy (90.53%) in 12 time-steps for CIFAR10, the proposed normalization method loses the energy benefits of SNNs and is similar to ANN in terms of the type of computation. The authors in [2]  propose a learning rule that requires computing loss at every time-step and the goal of the training is to teach output neurons to produce a desired firing sequence.  Therefore, for classification tasks the output neuron of the correct class is trained to spike at every time-step; this will increase the overall spiking activity of the network (4% of neurons spike at every time-step) and diminish the energy-efficiency.  We compare the results for the same architecture in the Table below and our approach performs better than both the networks for the same or less number of time-steps. Additionally, we show that our method is scalable to networks with residual connections and for larger datasets like CIFAR100, and ImageNet
>
> | Model     	| Method                                                	| Dataset  	| Architecture  	| Accuracy  	| Timesteps  	|
> |-----------	|-------------------------------------------------------	|----------	|---------------	|-----------	|------------	|
> | [1]       	| Surrogate gradient with normalization (requires MAC)  	| CIFAR10  	| CIFARNet      	| 90.53%    	| 12         	|
> | [2]       	| Inter-neuron and intra-neuron optimization            	| CIFAR10  	| CIFARNet      	| 91.41%    	| 5          	|
> | Our work  	| DIET-SNN                                              	| CIFAR10  	| CIFARNet      	| 91.59%    	| 5          	|
>
> CIFARNet: 128C3(encoding)-256C3-AP2-512C3-AP2-1024C3-512C3-1024FC-512FC-10
>
>
> [1] Wu, Y., Deng, L., Li, G., Zhu, J., Xie, Y., & Shi, L. (2019, July). Direct training for spiking neural networks: Faster, larger, better. In Proceedings of the AAAI Conference on Artificial Intelligence (Vol. 33, pp. 1311-1318).
>
> [2] Zhang, W., & Li, P. (2020). Temporal Spike Sequence Learning via Backpropagation for Deep Spiking Neural Networks. arXiv preprint arXiv:2002.10085.

---

> > ### Author Response · Authors · 2020-11-19
> > **Response to Reviewer #4 Part 2/2**
> >
> > **Reviewer's comment:**  To my understanding, the method proposed in this paper has nothing different compared to the existing BPTT method with the surrogate gradient. I hope the authors can claim clearly the novelty of this paper. For example, how is the proposed method different or better from [3] and [4]. As far as I know, the only difference is that the threshold and leaky parameters are also trained in this paper. However, this can also be easily done in other existing methods like [2][3][4]. There's no doubt their performance can also be improved since it introduces more tunable parameters.
> >
> > **Author's response:** The authors in [3,4] apply surrogate gradient training to train only the weights of the network. In this work, we employ a surrogate gradient to train weights, leak, and threshold jointly that improves the performance. The authors in [3,4] could have trained the leak and threshold but that was not shown in their works. The authors in [5] proposed a training method that employs weight sharing between ANN and SNN. They achieved competitive performance on CIFAR10 (90.98% with 8 time-steps), however, they achieved a top-1 accuracy of 50% on the ImageNet dataset with 10 time-steps, compared to 69% with 5 time-steps shown in our work. The authors in [6,7] employ threshold/leak training but the evaluation is only performed for simple datasets (MNIST). Therefore, the evaluation of deeper architectures and challenging datasets is non-trivial and requires significant engineering effort. In this work, we show the best state-of-the-art accuracies for challenging datasets (CIFAR10, CIFAR100, ImageNet) on VGG and ResNet architectures with a lower number of time-steps.
> >
> > **Reviewer's comment:** Tuning threshold and leaky parameters may be unbiologically plausible.
> >
> > **Author's response:** We agree that some of the concepts like end-to-end backpropagation and surrogate gradient training may not be bio-plausible. Also, the simple LIF neuron model may not be sufficient to represent the complex dynamics in the biological brain. However, research has shown that neurons in different regions of the biological brain operate with different leak [8] and threshold [9]. In this work, our goal is not to demonstrate bio-plausibility but to engineer a spiking network that can achieve high accuracy and better energy-efficiency.
> >
> > **Reviewer's comment:** What is the weight optimization method? As shown in Table 1, it seems the weight optimization contributes more to the performance than the proposed method.
> >
> > **Author's response:** The column 'weight optimization' in Table-1 is not a different method but refers to a spiking network trained with surrogate gradient to optimize only the weights. The weights are the major parameters in the network and therefore most of the performance depends on it. However, if we jointly train weights with other parameters (leak and threshold) the performance further improves as shown in the column 'DIET-SNN' of Table-1.
> >
> > [2] Zhang, W., & Li, P. (2020). Temporal Spike Sequence Learning via Backpropagation for Deep Spiking Neural Networks. arXiv preprint arXiv:2002.10085.
> >
> > [3] Wu, Y., Deng, L., Li, G., Zhu, J., & Shi, L. (2018). Spatio-temporal backpropagation for training high-performance spiking neural networks. Frontiers in neuroscience, 12, 331.
> >
> > [4] Shrestha, S. B., & Orchard, G. (2018). Slayer: Spike layer error reassignment in time. In Advances in Neural Information Processing Systems (pp. 1412-1421).
> >
> > [5] Jibin Wu et al., 2019, A Tandem Learning Rule for Effective Training and Rapid Inference of Deep Spiking Neural Networks
> >
> > [6] Bojian Yin et al. 2020, Effective and Efficient Computation with Multiple-timescale Spiking Recurrent Neural Networks
> >
> > [7] Fang W (2020) Leaky Integrate-and-Fire Spiking Neuron with Learnable Membrane Time Parameter.
> >
> > [8] Lu, Boxun, et al. "The neuronal channel NALCN contributes resting sodium permeability and is required for normal respiratory rhythm." Cell 129.2 (2007): 371-383.
> >
> > [9] Connors, Barry W., and Michael A. Long. "Electrical synapses in the mammalian brain." Annu. Rev. Neurosci. 27 (2004): 393-418.

---

### Official Review · AnonReviewer3 · 2020-10-28
**Interesting results for direct-trained low-latency SNN but with limited novelty and incomplete ablation studies**

**Rating:** 5
**Confidence:** 4

**Review:**

This paper proposes a Spiking Neural Network (SNN) training method that jointly optimizes input spike encoding parameters, spiking neuron parameters (membrane leak and voltage threshold), and weights in an end-to-end fashion using gradient descent. Compared to SNNs with only weight optimization, the SNN trained by the proposed method significantly decreases the inference latency and results in high activation sparsity with minimal accuracy decrease.

Highlights:

1. Energy-efficiency is currently the main advantage of SNN. Two primary factors that determine the energy-efficiency of an SNN are inference latency and activation sparsity. The method proposed in the paper directly targets both factors, and the intuitions are very clear.

2. The results show the joint training approach directly contributes to the improvement of SNN's inference latency and activation sparsity (as shown in Fig. 2 and Fig. 3 in the paper). It's also interesting to see the method can learn low membrane leak and high voltage threshold at the same time for later layers of VGG16, which significantly improve activation sparsity.

3. The paper proposes a new spike generation function for LIF neurons in equation (2) of section 3. The function places the voltage threshold in the computational graph of backpropagation, making training voltage threshold possible in an elegant way.

Concerns:

1. The reviewer’s main concern with the paper is its limited novelty. The joint end-to-end training of neuron parameters has already been proposed in a recent paper ([Bojian Yin et al. 2020]). Moreover, in addition to gradient-based training, there are also local learning methods like intrinsic plasticity that tune the neuron parameters based on the activity of the SNN (for example, [Wenrui Zhang et al. 2019], [Anguo Zhang et al. 2019]). The paper needs to better position the proposed method with respect to these existing related works.

2. Table 3 shows the computation overheads as a result of the direct encoding layer is around 1%. However, the large number of encoded spiking activities can slow down the inference speed when the SNN is deployed on currently available neuromorphic chips (for example, Intel's Loihi or IBM's TrueNorth) since these chips lack an efficient solution for injecting external spikes. It will be great if the paper can show how the DIET-SNN performs with different numbers of convolution channels in the encoding layer and examine if using fewer input neurons for encoding will hurt the accuracy.

3. The proposed method uses backpropagation through time (BPTT) to train the SNN. Many recent papers that train SNN using spike-train level learning methods also achieve low latency and sparse synaptic activities (for example, [Jibin Wu et al., 2019], [Yingyezhe Jin et al., 2018]). The paper lacks experiments for comparing the performance, inference latency, and activation sparsity with these existing methods.

4. The ablation study in section 6 uses IF neurons for the first 3 experiments and LIF neurons for the last experiment which seems unfair. The ablation study should show that DIET-SNN can do better than an SNN using LIF neurons with fixed (preset) membrane leak and voltage threshold. The current ablation study cannot rule out the possibility that an SNN with fixed (preset) low membrane leak and high voltage threshold performs as well as or even better than DIET-SNN in terms of accuracy and activation sparsity.

5. The ablation study in section 6 shows the performance of only one inference timesteps value for each SNN. To show a complete picture of how the performance changes with the latency, the paper needs to train each SNN with the same group of latency values (for example, all 4 SNN trained with 150, 35, 25, and 20 timesteps).

6. In section 6, the paper claims that optimizing the threshold has similar consequences as optimizing the weights. The co-optimization of them can lead to faster convergence and lower latency. However, there are no experiments to support this argument. The reviewer suggests conducting additional experiments for this.


I want to thank the author for addressing my concerns. Many of my concerns are resolved. I have updated my rating after the discussion. I agree with the authors that the novelty of the submission lies in scaling up the existing methods. However, I'm not sure if this is enough for the paper's scientific significance required by this conference.


Bojian Yin et al. 2020, Effective and Efficient Computation with Multiple-timescale Spiking Recurrent Neural Networks

Wenrui Zhang et al. 2019, Information-theoretic intrinsic plasticity for online unsupervised learning in spiking neural networks

Auguo Zhang et al. 2019, Fast and robust learning in Spiking Feed-forward Neural Networks based on Intrinsic Plasticity mechanism

 Jibin Wu et al., 2019, A Tandem Learning Rule for Effective Training and Rapid Inference of Deep Spiking Neural Networks

Yingyezhe Jin et al., 2018, Hybrid macro/micro level backpropagation for training deep spiking neural networks

---

> ### Author Response · Authors · 2020-11-19
> **Response to Reviewer #3 Part 1/3**
>
> **Reviewer's comment:** The reviewer’s main concern with the paper is its limited novelty. The joint end-to-end training of neuron parameters has already been proposed in a recent paper ([Bojian Yin et al. 2020]). Moreover, in addition to gradient-based training, there are also local learning methods like intrinsic plasticity that tune the neuron parameters based on the activity of the SNN (for example, [Wenrui Zhang et al. 2019], [Anguo Zhang et al. 2019]). The paper needs to better position the proposed method with respect to these existing related works.
>
> **Author's response:** Thank you for your comments. We wrote a common reply on the top regarding the novelty of the work. Bojian Yin et al. proposed a spiking recurrent neural network (SRNN) with an adaptive threshold (threshold changes with time) and trained with end-to-end backpropagation-through-time (BPTT). Though they optimize neuron parameters with gradient-descent, our work significantly differs in the following ways: 1. The LIF equations used in [1] are more complex and have more parameters compared to our work. This increases the memory requirement and the associated energy in fetching and storing the parameters. 2. We employ feedforward deep (>16 layers) spiking convolutional network compared to shallow(<3 layers) spiking recurrent networks used in [1]. 3. For static image datasets, the authors in [1] process the image pixels as sequence and therefore the number of timesteps is proportional to the image size. In our work, we process the images with convolutional layers that are more suited for image datasets. The authors in [1] report results for only MNIST images, whereas in this work we show the scalability of this approach on larger and challenging datasets like CIFAR100 and ImageNet.
>
> The authors in [2,3] proposed an intrinsic plasticity (IP) rule to update the leaky resistance and the time constant of membrane potential in LIF neurons. The authors in [2] do not specifically talk about any method to update the weights, but in the experiments, they used a previously proposed spike-based learning rule. They show the efficacy of their approach by training a liquid state machine (LSM) where the proposed IP method is applied only on the reservoir neurons. Therefore, it is not clear how this method will perform in a network with multiple layers. They report their results only for the TI46 speech corpus and CitySpace dataset and therefore we cannot compare the results directly with our work. The authors in [3] perform ANN-SNN conversion and then employ the IP method to update neuron parameters. The ANN is trained with end-to-end backpropagation and the local IP rule is only applied to update the neuron parameters. Here again, the experiments were limited to the MNIST dataset with fully-connected networks. The intrinsic plasticity rules and other local learning rules proposed for SNNs perform well on simple tasks (MNIST) but do not scale to challenging datasets (CIFAR100, ImageNet). Though local learning rules are desirable for online learning in neuromorphic hardware, they still do not perform as well as the backpropagation based methods. In our work, the goal is to achieve state-of-the-art accuracy on difficult tasks with minimum inference latency and energy. Therefore, we adopt the end-to-end backpropagation method and report better performance on challenging datasets with low inference latency.
>
> [1] Bojian Yin et al. 2020, Effective and Efficient Computation with Multiple-timescale Spiking Recurrent Neural Networks
>
> [2] Wenrui Zhang et al. 2019, Information-theoretic intrinsic plasticity for online unsupervised learning in spiking neural networks
>
> [3] Auguo Zhang et al. 2019, Fast and robust learning in Spiking Feed-forward Neural Networks based on Intrinsic Plasticity mechanism
>
> [4] Jibin Wu et al., 2019, A Tandem Learning Rule for Effective Training and Rapid Inference of Deep Spiking Neural Networks
>
> [5] Yingyezhe Jin et al., 2018, Hybrid macro/micro level backpropagation for training deep spiking neural networks

---

> > ### Author Response · Authors · 2020-11-19
> > **Response to Reviewer #3 Part 2/3**
> >
> > **Reviewer's comment:** Table 3 shows the computation overheads as a result of the direct encoding layer is around 1%. However, the large number of encoded spiking activities can slow down the inference speed when the SNN is deployed on currently available neuromorphic chips (for example, Intel's Loihi or IBM's TrueNorth) since these chips lack an efficient solution for injecting external spikes. It will be great if the paper can show how the DIET-SNN performs with different numbers of convolution channels in the encoding layer and examine if using fewer input neurons for encoding will hurt the accuracy.
> >
> > **Author's response:**  Thank you for bringing up this important point of injecting external spikes in current neuromorphic chips. We trained a network with CIFARNet architecture (XC3(encoding)-256C3-AP2-512C3-AP2-1024C3-512C3-1024FC-512FC-10) for the CIFAR10 dataset with a varying number of channels (X) in the encoding layer and the results are shown in Table below. The authors in [6] used the PCA method to represent the image in a lower dimension before converting it to spikes; the same can be applied in our training methodology as well along with reducing the encoding channels.
> >
> > | #encoding channels 	| Accuracy 	| Time-steps 	|
> > |-----------------------	|----------	|------------	|
> > | 4                     	| 90.14%   	| 5          	|
> > | 8                     	| 90.92%   	| 5          	|
> > | 16                    	| 91.06%   	| 5          	|
> > | 32                    	| 91.52%   	| 5          	|
> > | 64                    	| 91.69%   	| 5          	|
> > | 128                   	| 91.59%   	| 5          	|
> >
> > **Reviewer's comment:** The proposed method uses backpropagation through time (BPTT) to train the SNN. Many recent papers that train SNN using spike-train level learning methods also achieve low latency and sparse synaptic activities (for example, [Jibin Wu et al., 2019], [Yingyezhe Jin et al., 2018]). The paper lacks experiments for comparing the performance, inference latency, and activation sparsity with these existing methods.
> >
> > **Author's response:** Thank you for pointing this out. We provide a table below to compare our results with the mentioned work and have also updated the paper (Table 2) to include this comparison. The authors in [4] propose a method to train two networks (ANN and SNN) simultaneously (doubling the overall memory requirement) and share the weights between them. The weights are trained in ANN and copied to SNN; the activations in ANN are computed as the sum of spikes in SNN for that layer. As the training is not performed in the spiking domain, the temporal information is not utilized and the method fails to achieve competitive accuracy on challenging tasks (ImageNet).Backpropagation through time (BPTT) performs considerably better than the spike-train level training methods as shown below. Also, spike-train level methods do not scale well to larger datasets like ImageNet. The authors in [5] showed results for only the MNIST dataset.
> >
> > | Model                   	| Method                       	| Dataset   	| Architecture  	| Accuracy  	| Timesteps  	| Spike Rate  	|
> > |-------------------------	|------------------------------	|-----------	|---------------	|-----------	|------------	|-------------	|
> > |  Jibin Wu et al., 2019  	| ANN-SNN weight sharing       	| CIFAR10   	| CifarNet      	| 90.98     	| 8          	| 0.32        	|
> > | Our work                	| DIET-SNN                     	| CIFAR10   	| VGG16         	| 92.70     	| 5          	| 0.39        	|
> > | Jibin Wu et al., 2019   	| ANN-SNN weight sharing  	| ImageNet  	| AlexNet       	| 50.22     	| 10         	| -           	|
> > | Our work                	| DIET-SNN                     	| ImageNet  	| VGG16         	| 69.00     	| 5          	| 0.41        	|
> >
> > [4] Jibin Wu et al., 2019, A Tandem Learning Rule for Effective Training and Rapid Inference of Deep Spiking Neural Networks
> >
> > [5] Yingyezhe Jin et al., 2018, Hybrid macro/micro level backpropagation for training deep spiking neural networks
> >
> > [6] Frady, E. Paxon, et al. "Neuromorphic Nearest-Neighbor Search Using Intel's Pohoiki Springs." arXiv preprint arXiv:2004.12691 (2020).

---

> > > ### Author Response · Authors · 2020-11-19
> > > **Response to Reviewer #3 Part 3/3**
> > >
> > > **Reviewer's comment:** The ablation study in section 6 uses IF neurons for the first 3 experiments and LIF neurons for the last experiment which seems unfair. The ablation study should show that DIET-SNN can do better than an SNN using LIF neurons with fixed (preset) membrane leak and voltage threshold. The current ablation study cannot rule out the possibility that an SNN with fixed (preset) low membrane leak and high voltage threshold performs as well as or even better than DIET-SNN in terms of accuracy and activation sparsity.
> > >
> > > **Author's response:** Thank you for pointing this out. We did perform experiments with a fixed leak (0.99 and 0.95) for the first 3 networks (a, b, c) in the ablation study but that lead to lower accuracy for the same number of time-steps. We achieved an accuracy of 91.72%, 91.84%, 91.76% for networks (a), (b), and (c), respectively. The same leak value for all layers is not the optimal setting and therefore the network without leak (leak=1) performed better. Reducing the leak below 0.95 degraded the accuracy further.
> > >
> > > **Reviewer's comment:** The ablation study in section 6 shows the performance of only one inference timesteps value for each SNN. To show a complete picture of how the performance changes with the latency, the paper needs to train each SNN with the same group of latency values (for example, all 4 SNN trained with 150, 35, 25, and 20 timesteps).
> > >
> > > **Author's response:**  The ablation study was performed under iso-accuracy. So, for each network, we selected the minimum number of time-steps needed to achieve >92% accuracy for the CIFAR10 dataset on VGG16 architecture. The network (a) will have an accuracy much less than 92% if evaluated under 35 time-steps or lower. Similarly, the networks (b), (c), and (d) will not have a significant increase in accuracy if trained for 150 time-steps. The high time-steps requirement comes from Poisson encoding and the goal in SNNs is to always find a setting with a lower number of timesteps.
> > >
> > > **Reviewer's comment:** In section 6, the paper claims that optimizing the threshold has similar consequences as optimizing the weights. The co-optimization of them can lead to faster convergence and lower latency. However, there are no experiments to support this argument. The reviewer suggests conducting additional experiments for this.
> > >
> > > **Author's response:**  We apologize if the writing of Section 6 is not clear to express this distinction. In Section 6 the networks (b) and (c) represent the two cases. For the network (b) only the weights are trained while the thresholds are held constant and the network achieves convergence in 35 time-steps (25 time-steps in updated results). Whereas for the network (c), both the weights and thresholds are optimized jointly and the network achieves convergence in 25 time-steps (15 time-steps in updated results). Therefore, co-optimization reduces inference latency by 10 time-steps.

---

### Author Response · Authors · 2020-11-19
**General Response**

We thank all the reviewers for their insightful comments and feedback. We are encouraged that the reviewers liked the elegant way of introducing threshold voltage in the computational graph for gradient computation (R3), the demonstration on challenging datasets like CIFAR100 and ImageNet (R2, R4) and highlight that this is the first work to claim better accuracy on these datasets with fewer time-steps (R4). The reviewers also raised some concerns that we have addressed below individually. In the meantime, we made minor changes in the source code that reduce the latency to 5 timesteps for all datasets and architectures. The order of execution of the equations in the code was modified. Initially, we were performing the thresholding function based on the potential in the previous time-step, but now we changed it to the potential in the current time-step. The computation based on the previous time-step added a delay of 1 time-step that accumulated for each layer, so for a network with 16 layers, it introduced an additional delay of 16 time-steps. We have updated Table 1,2,3 and Fig. 2,3 to reflect the new results. The updated source code is available in the supplementary material.

---

> ### Author Response · Authors · 2020-11-19
> **Novelty of the work**
>
> The reviewers pointed out that some of the concepts also appear in other works recently (within six months) published [1,2,3,4]. Note, however, in this work, we combine the concepts of hybrid training, direct analog coding, threshold optimization, and leak optimization to evaluate challenging datasets (CIFAR100, ImageNet) and achieve state-of-the-art performance on deep VGG and ResNet architectures that have not been shown in previous works [1-9]. Most of the other works have only focused on AlexNet or shallower architectures without residual connections. In SNNs, adding deeper layers introduces the problem of diminishing spikes in the output layers. The spikes in SNNs travel from the input layer to the output layer. In each layer, the weighted sum of incoming spikes is accumulated and then compared to a threshold to generate output spikes. To maintain sufficient spike activity in deeper layers, the spike rate in the input layer needs to be decently high, and the thresholds in subsequent layers need to be appropriately tuned. The authors in [3] achieved a top-1 accuracy of 50% on the ImageNet dataset with 10 time-steps, compared to 69% with 5 time-steps shown in our work. Therefore, the evaluation of deeper architectures and challenging datasets is non-trivial and requires significant engineering effort. Some of the spike-train level training methods [2,3,6], and intrinsic plasticity-based learning rules [5] achieve low latency on MNIST/CIFAR10 but fail to achieve competitive performance on complex tasks. The surrogate gradient methods [8,9] fails to show competitive performance on larger datasets with low latency. The end-to-end backpropagation method used in our work achieves state-of-the-art accuracies on both simple and challenging tasks with low latency. The training methods proposed in [2,3,7] achieve low inference latency (5-12 time-steps) for the CIFAR10 dataset. However, the authors in [2]  propose a learning rule that requires computing loss at every time-step, and the goal of the training is to teach output neurons to produce a desired firing sequence.  Therefore, for classification tasks the output neuron of the correct class is trained to spike at every time-step; this will increase the overall spiking activity of the network (4% of neurons spike at every time-step) and diminish the energy-efficiency. The authors in [3] propose a method to train two networks (ANN and SNN) simultaneously (doubling the overall memory requirement) and share the weights between them. The weights are trained in ANN and copied to SNN; the activations in ANN are computed as the sum of spikes in SNN for that layer. As the training is not performed in the spiking domain, the temporal information is not utilized and the method fails to achieve competitive accuracy on challenging tasks (ImageNet). The authors in [7] proposed a normalization method (NeuNorm) that computes a weighted summation of spike count and uses that quantity as the input to the convolutional layer instead of the raw spike signals. Therefore, the convolution requires the multiply-and-accumulate (MAC) operation as both the input and the weight are real-valued quantities. In SNN, the major advantage is that the expensive MAC operation (needed in ANN) is reduced to simple additions due to binary inputs. Although the authors achieved competitive accuracy (90.53%) in 12 time-steps for CIFAR10, the proposed normalization method loses the energy benefits of SNNs and is similar to ANN in terms of the type of computation. In this work, we achieve high accuracy, high sparsity in spikes, and low energy by employing binary spike communication with a lower number of timesteps. We have also open-sourced the code and will make the trained models available for other researchers to build upon it. We believe that the competitive performance of SNNs on challenging datasets makes it more attractive for energy-efficient machine learning.
>
> [1] Bojian Yin et al. (ICONS 2020) Effective and Efficient Computation with Multiple-timescale Spiking Recurrent Neural Networks
>
> [2] Wenrui Zhang et al. (accepted in Neurips 2020) Temporal Spike Sequence Learning via Backpropagation for Deep Spiking Neural Networks.
>
> [3] Jibin Wu et al. (arxiv 2020) A Tandem Learning Rule for Effective Training and Rapid Inference of Spiking Neural Networks
>
> [4] Wei Fang (arxiv 2020), Leaky Integrate-and-Fire Spiking Neuron with Learnable Membrane Time Parameter
>
> [5] Auguo Zhang et al. (2019), Fast and robust learning in Spiking Feed-forward Neural Networks based on Intrinsic Plasticity mechanism
>
> [6] Yingyezhe Jin et al. (2018), Hybrid macro/micro level backpropagation for training deep spiking neural networks
>
> [7] Yujie Wu et al. (2018), Direct training for spiking neural networks: Faster, larger, better
>
> [8] Yujie Wu et al. (2018), Spatio-temporal backpropagation for training high-performance spiking neural networks
>
> [9] Sumit Bam Shrestha et al. (2018), Slayer: Spike layer error reassignment

---

### Decision · Program_Chairs · 2021-01-07
**Final Decision**

**Decision:**

Reject

**Comment:**

The manuscript presents a training method for Spiking Neural Networks (SNN). The method jointly optimizes input spike encoding parameters, spiking neuron parameters (membrane leak and voltage threshold), and weights in an end-to-end fashion using gradient descent. SNNs are very interesting for energy-efficient implementations of neural networks. Their energy efficiency strongly depends on inference latency (SNNs compute in time, unlike feed-forward ANNs) and activation sparsity.

All reviewers acknowledged that the approach directly improves inference latency and activation sparsity on large convolutional models at very good performance levels.

The main concern of all reviewers was the limited conceptual novelty. The paper combines some known techniques (hybrid SNN training, direct input encoding, training of neuron parameters like leak time constant and threshold) and scale the setup up to large networks and datasets (e.g. ImageNet).

In summary, the paper presents impressive results, but the conceptual innovation is missing.